# CSF1R Inhibition Combined with GM-CSF Reprograms Macrophages and Disrupts Protumoral Interplays with AML Cells

**DOI:** 10.3390/cancers13215289

**Published:** 2021-10-21

**Authors:** Tatiana Smirnova, Caroline Spertini, Olivier Spertini

**Affiliations:** Service and Central Laboratory of Hematology, Centre Hospitalier Universitaire Vaudois, University of Lausanne, 1011 Lausanne, Switzerland; caroline.spertini@chuv.ch

**Keywords:** AML, macrophages, M-CSF, CSF1 receptor, GM-CSF, CD163, microenvironment, orientation, drug resistance, cytokines

## Abstract

**Simple Summary:**

Acute myeloid leukemia is a blood disease whose long-term treatment is not satisfactory due to frequent (>50%) relapse despite complete initial remission. By adopting protumoral properties, macrophages in the leukemia cell microenvironment may promote resistance to treatment leading to relapse. Our goal was to assess whether macrophages in contact with leukemia cells have an impact on blast survival and sensitivity to chemotherapy. We observed that leukemia cells can educate macrophages to support their survival and proliferation and reduce their sensitivity to therapy, as long as the CSF1 receptor remains active and they are in close contact. By inhibiting the activity of the CSF1 receptor, in the presence of GM-CSF, we could modify the macrophage phenotype and increase blast apoptosis and sensitivity to treatment. Our results indicate that leukemia therapies should not only target blasts but also their microenvironment and specifically macrophages and their receptor for CSF1.

**Abstract:**

Relapse is a major issue in acute myeloid leukemia (AML) and while the contribution of gene mutations in developing drug resistance is well established, little is known on the role of macrophages (MΦs) in an AML cell microenvironment. We examined whether myeloblasts could educate MΦs to adopt a protumoral orientation supporting myeloblast survival and resistance to therapy. Flow cytometry analyses demonstrated that M2-like CD163^+^ MΦs are abundantly present, at diagnosis, in the bone marrow of AML patients. We showed that myeloblasts, or their conditioned medium, polarize monocytes to M2-like CD163^+^ MΦs, induce the secretion of many protumoral factors, and promote myeloblast survival and proliferation as long as close intercellular contacts are maintained. Importantly, pharmacologic inhibition of the CSF1 receptor (CSF1R), in the presence of GM-CSF, reprogrammed MΦ polarization to an M1-like orientation, induced the secretion of soluble factors with antitumoral activities, reduced protumoral agonists, and promoted the apoptosis of myeloblasts interacting with MΦs. Furthermore, myeloblasts, which became resistant to venetoclax or midostaurin during their interplay with protumoral CD163^+^ MΦs, regained sensitivity to these targeted therapies following CSF1R inhibition in the presence of GM-CSF. These data reveal a crucial role of CD163^+^ MΦ interactions with myeloblasts that promote myeloblast survival and identify CSF1R inhibition as a novel target for AML therapy.

## 1. Introduction

AML is a very aggressive disease with poor long-term survival [1]. Seventy to ninety percent of adult patients under the age of 60 achieve complete remission; however, 50–70%, with intermediate or high genetic risk, relapse within 5 years. The European LeukemiaNet (ELN) 2017 recommendations for AML treatment are presently based on risk stratification by genetics and also consider the possibility to target the AML microenvironment as novel therapies [2].

Chemotherapy and hematopoietic stem cell transplantation most often allow the achievement of sustained complete remission. Moreover, targeted therapies improve the AML response to chemotherapy and favorably affect survival [3]. However, they remain restricted to a small number of AML categories. Over time and regardless of the treatment, the tumor microenvironment strongly contributes to promoting the growth and survival of drug-resistant subclones, leading to relapse [4,5]. Several clinical studies have demonstrated that heavy tumor infiltration by MΦs is of poor prognosis: tumor-associated macrophages (TAMs) promote tumor cell growth, resistance to treatment, and escape from immune surveillance [4,6].

TAMs are often called M2-like protumoral MΦs while the classical M1-like MΦs have inflammatory, antitumor properties [7,8]. However, MΦ plasticity is complex [9,10], and the M1/M2 nomenclature does not correspond to observations made in tumors, where mixed phenotype MΦs, expressing both M1 and M2 markers, are observed. In the tumor microenvironment, malignant cells activate and educate MΦs, which secrete cytokines and promote angiogenesis, matrix remodeling, metastasis, immunosuppression, cancer cell survival, and drug resistance [9,11,12,13].

Tumor escape from immune surveillance is further promoted by the expression of the “do not eat me” signal, CD47 [11,14] and the increased expression of programmed cell death protein-1 (PD1) by tumor-infiltrating T-cells and of its ligand PD-L1 (programmed death-ligand 1) by TAMs [15]. While the involvement of TAMs in the solid cancer microenvironment is established [16,17], little information is available on their role in human AML. In mice, heterogeneity in MΦ polarization was observed in an MLL-AF9 AML model [18,19]. Myeloblasts can recruit MΦs to support their proliferation in bone marrow (BM) and spleen [20].

The scavenger receptor CD163 is a marker of TAMs with protumoral characteristics [7,12,13]. Several studies reported that CD163 is a poor prognosis marker in solid tumors [13] and in AML [21]. CD163 expression appears to be required to allow mouse and human MΦs to adopt a protumoral phenotype and support malignant cell proliferation. Thus, in a mouse model of sarcoma, tumor growth was abrogated in CD163 deficient mice [22]. CD163-expressing TAMs may contribute to create an immunosuppressive microenvironment. In melanoma, the depletion of CD163^+^ MΦs promotes tumor infiltration by activated T lymphocytes and tumor regression [7].

Targeting TAM polarization is a novel important therapeutic approach in aggressive cancers [16,23]. The colony-stimulating factor-1 (CSF-1, also called M-CSF), which controls the survival, proliferation, differentiation, and polarization of MΦs by binding to its receptor, the CSF1R, is a promising therapeutic target in aggressive tumors and possibly in AML [24]. In multiple myeloma and mantle cell lymphoma, the inhibition of CSF1R can reprogram TAMs and reverse drug-resistance [25,26,27,28]. However, whether MΦs contribute to AML cell proliferation and/or survival in creating a protumorigenic niche associated with a protumoral MΦ orientation is not established.

Using human AML cell lines and primary myeloblasts, we examined the impact of AML cells on allogeneic and autologous MΦs. We showed that the co-culture of primary myeloblasts with healthy donor (HD) monocytes or with BM cells in their microenvironment induces a protumoral MΦ orientation (AML-MΦ), which supports blast cell proliferation and survival. Furthermore, using CSF1R inhibitors and granulocyte-macrophage colony-stimulating factor (GM-CSF), we succeeded in reprogramming protumoral AML-MΦs and in reversing drug-resistance induced by myeloblast cross-talk with MΦs.

## 2. Materials and Methods

### 2.1. Patient Samples

Heparinized blood and BM samples were obtained from 42 newly diagnosed AML patients at CHUV (Centre Hospitalier Universitaire Vaudois, Lausanne, Switzerland). Informed consent was obtained from all subjects involved in the study. After red cell lysis with ammonium chloride, the samples were processed for leukocyte immunostaining and analysis by flow cytometry (FC) [29]. Leukemia diagnosis was based on WHO 2016 classification [30] and ELN 2017 risk stratification by genetics [2] and are indicated in Appendix A. The study was conducted in accordance with the Declaration of Helsinki and approved by the Commission Cantonale d’Ethique sur la recherche sur l’être humain CER-VD (protocol 2017-01509, 9 November 2017).

### 2.2. Immunophenotypic Analyzes by Flow Cytometry

Cell proliferation, apoptosis, and immunophenotypic analyzes were performed using a Beckman Coulter Gallios flow cytometer. Patient samples were analyzed with a 10-color/24-antibody panel [29] according to ELN recommendations [31]. MΦs were detached from culture dishes in phosphate-buffered saline (PBS) containing 10 mM ethylenediaminetetraacetic acid (EDTA), for 10 min at 37 °C, washed with Roswell Park Memorial Institute (RPMI) 1640 and characterized using CD45, CD14, CD163, CD80, CD206, and CSF1R; additional blast and lymphocyte markers were used for patient BM and peripheral blood (PB) sample and BM co-culture analyses (Appendix A).

### 2.3. Healthy Donor-Derived Macrophages

The monocytic fraction was isolated by centrifuging HD PB anticoagulated with citrate phosphate dextrose-adenine 1 and diluted 1/2 with PBS for 10 min at 1000 g to prepare a buffy coat. The concentrated leukocytic fraction was then centrifuged at 400 g for 40 min on a Ficoll-Paque PLUS (GE Healthcare, Dietikon, Switzerland) gradient. Mononuclear cells were collected and plated in RPMI + 10% heat-inactivated fetal bovine serum (FBS) = plain medium (PM), see Appendix A for the schematic protocol; monocyte-derived MΦs exposed to 10 ng/mL recombinant human M-CSF (ImmunoTools, Friesoythe, Germany) were called M-MΦs and those exposed to 1000 U/mL GM-CSF (Miltenyi Biotec, Solothurn, Switzerland): GM-MΦs [25,32].

After overnight culture, the wells were rinsed to deplete non-adherent cells; >95% of adherent cells were CD14^+^ monocytes. The medium was changed, with the added factors, every 2 days, for 7–9 days, when CD163 expression was checked by FC. Alternatively, to obtain HD^AML^-MΦs, HD monocytes were stimulated for 7–9 days with (1) primary myeloblast or cell line conditioned medium (CM) diluted 1:1 with PM; with (2) AML cell lines (1–5 × 10^4^/well) or primary blasts (0.5–1 × 10^6^/well), co-cultured in direct contact with HD monocytes; or (3) separated from MΦs by a 0.4 µm-pore membrane insert (Transwell (TW), Falcon, Dietikon, Switzerland).

For MΦ reprogramming experiments, HD MΦ monolayers initially activated with M-CSF (M-MΦ) or blast CM (HD^AML^-MΦ) were washed with PBS and cultured for 7–9 more days in PM containing GM-CSF and GW2580 (1 μM, Sigma-Aldrich, Buchs, Switzerland) [25] or PLX3397 (100 nM, MedChemExpress, Luzern, Switzerland), renewed every 2 days. The change in MΦ CD163 expression after reprogramming was assessed by FC, as described above.

### 2.4. Cell Lines and Cell Culture

HL-60 (AML with maturation), U937 (monoblastic leukemia), NB4 (acute promyelocytic leukemia), OCI-AML3 (NPM1 mutated) cell lines were cultured in plain medium (PM) [29] and MV-4-11 (monoblastic AML with FLT3-ITD mutation) in Iscove’s Modified Dulbecco Medium + 10% FBS.

### 2.5. PKH Labeling and Apoptosis Detection

Cells were labeled with 10 µM PKH26 (Sigma-Aldrich) according to the manufacturer’s protocol and immediately added to rinsed wells containing adherent MΦs for assays. A fraction of the labeled cells were fixed in 1% paraformaldehyde and used to calculate the normalized PKH mean fluorescence intensity (MFI), as PKH MFI at day 4 divided by PKH MFI at day 0. Blast cell apoptosis was assessed by FC, analyzing cells stained with annexin V (AnnV, eBioscience, Zug, Switzerland).

### 2.6. Cleaved Caspase-3 Detection

Cells were cultured in 100% PM or in 75% PM supplemented with 25% MΦ-CM. After 48 h, the cells were lysed, and the cell lysates were subjected to SDS-PAGE, transferred to nitrocellulose membranes, immunoblotted with anti-cleaved caspase-3 antibody (Cell Signaling Technology) followed by horseradish peroxidase-linked goat anti-rabbit immunoglobulin G (IgG) antibody (Cell Signaling Technology), and revealed with Luminata Forte chemiluminescence (Millipore, Schaffhausen, Switzerland) [29]. As a loading control, membranes were also immunoblotted with anti-α-tubulin antibody (Sigma-Aldrich) followed by horseradish peroxidase-conjugated sheep anti-mouse IgG antibody (Amersham, Dietikon, Switzerland) and revealed with Luminata Crescendo chemiluminescence (Millipore). Blocking antibodies were from InvivoGen (anti-IL-6) and R&D Systems (anti-tumor necrosis factor α (anti-TNF-α)).

### 2.7. Cytokine Arrays

The CM of the indicated leukemia cell lines, primary myeloblasts or MΦs were collected and centrifuged twice at 1500 rpm for 10 min, twice for 10 min at 2500 rpm, and filtered with 0.22 μm low protein binding syringe filter. CM was collected at day 7, from M- or GM-MΦs cultured in medium containing M- or GM-CSF, respectively. CM from reprogrammed MΦs (R^GM/GW^-MΦs) was obtained after one week of culture with M-CSF, followed by another week in medium supplemented with GM-CSF and GW2580. CM from M-, GM-, and R^GM/GW^-MΦs were obtained from at least three different HD and pooled for cytokine assays. The CM from primary AML blasts (mononuclear cell suspension containing >95% blast cells) co-cultured with MΦ were also analyzed and processed as above.

Cytokine array assays were conducted according to the manufacturer’s protocol (R&D Systems, Human XL Cytokine Array kit #ARY022B). X-ray films were digitalized with ImageScanner III (GE Healthcare) and dot pixel density was measured with ImageQuant software (Molecular Dynamics, Glattbrugg, Switzerland). The mean of duplicates was normalized to the mean pixel density of the six reference spots.

### 2.8. Myeloblast Resistance to Venetoclax and Midostaurin

HD MΦs were polarized for 7 days with M-CSF and then either reprogrammed for 7 more days in medium containing GW2580 or PLX3397 and GM-CSF or kept in M-CSF-containing medium. MΦ orientation and reprogramming was confirmed by FC. Medium was then removed, and MΦs were washed with PBS. Following this, 150 × 10^3^ NB4 or MV-4-11 were added to MΦ monolayers in PM with inhibitors (venetoclax from LC Laboratories, or midostaurin from Sigma-Aldrich). After 48 h, cells were removed from the wells, stained for AnnV and/or 7-aminoactinomycin D (7-AAD), and analyzed by FC. Experiments were repeated using MΦs from at least three different HD.

### 2.9. Statistical Analysis

Statistical significance of differences between groups was examined with the Mann-Whitney (M-W) test between two nonparametric unpaired groups, the paired *t*-test between parametric matched samples, or the Wilcoxon matched-pairs signed rank (WMP) test between nonparametric matched samples. *p* values < 0.05 were considered as significant.

## 3. Results

### 3.1. Presence of CD163-Expressing Macrophages in BM from AML Patients

BM and/or PB were obtained from 42 adult patients with newly diagnosed AML (Appendix A). MΦs and myeloblasts were identified by multiparameter FC with a panel of antibodies (Appendix A, see Methods for details) [29,31]. BM and PB analyses showed that independently of ELN risk categories [2], the majority of MΦs in AML patients expressed CD163 (Figure 1A), a marker frequently associated with a protumoral phenotype [18,19,20,21]. By contrast, CD80, a marker typically associated with the inflammatory phenotype and whose expression and functions remain poorly characterized in AML, was expressed at much lower frequency [25,27].

### 3.2. Myeloblasts Polarize HD Monocytes to Macrophages

We first monitored the differentiation of HD Mo into MΦs by adding either M- or GM-CSF in PM for one week. Monocytes cultivated in M-CSF differentiated into M2-like CD163^high^ MΦs, whereas those cultured in GM-CSF differentiated into M1-like CD163^low^ MΦs (Figure 1B). To determine whether myeloblasts can polarize protumoral MΦs and induce CD163 expression at their surface, we performed co-cultures of HD Mo with human AML cell lines or primary myeloblasts obtained from AML patients. After 7 days, HD Mo differentiated into MΦs (HD^AML^-MΦs; Figure 1C, contact), whose CD163 expression was similar to that of HD MΦs cultured in medium supplemented with M-CSF.

To determine if direct contact is required for myeloblasts to polarize monocytes into MΦs, they were separated with a 0.4 μm-pore membrane-insert (TW). CD163 expression on HD^AML^-MΦs was consistently observed even in the absence of direct contact (Figure 1C). Similar results were obtained by stimulating monocytes with CM from AML cell lines or primary myeloblasts. As control, the co-culture of normal hematopoietic stem progenitor cells (HSPC) with HD Mo, under the same conditions as AML cells, did not induce CD163 expression at the surface of monocytes (Appendix A). In addition, culture in PM or with other inflammatory factors (lipopolysaccharide, TNF-α, IL-1β) did not induce CD163 expression; moreover, boiled U937 CM also failed to induce CD163 expression (Appendix A). These results suggest that soluble factors released by myeloblasts contribute to inducing MΦ polarization.

Several other polarization markers were analyzed: the expression level of CD80 was expectedly higher in GM- than in M-MΦs (Appendix A) but exhibited some variability among HD^AML^-MΦs (Appendix A). M- and GM-MΦs exhibited variable expression of CD206 (Appendix A), a marker typically associated with protumoral MΦ activation in different tumor types including AML [33]. CD206 was also expressed by most HD^AML^-MΦs (Appendix A), both after contact with blasts or with their CM only.

As M-CSF plays a crucial role in polarizing M-MΦs, we assessed whether inhibiting its receptor could prevent the up-regulation of CD163 expression induced by myeloblasts CM. Indeed, GW2580 and PLX3397, when added into CM, significantly prevented CD163 upregulation (Figure 1D), confirming a major role for CSF1R signaling in inducing CD163 expression.

To understand what factors in AML may be important in contributing to alternative MΦ polarization and the induction of CD163 expression, we performed semi-quantitative cytokine array analyses using myeloblast CM from three AML patients with favorable, intermediate, or high genetic risk (patients #3, #18, and #23, respectively) and from two AML cell lines (HL-60 and U937; Figure 1E). The assay can detect 105 factors, indicated in Appendix A, which shows the pixel density of dots revealed on membranes illustrated in Appendix A), some of which may contribute to promote cancer growth, cell migration, angiogenesis, and metastasis, such as CXCL5, the urokinase-type plasminogen activator receptor (uPAR), the matrix metalloproteinase 9 (MMP-9), serpin E1 [34], and the growth differentiation factor-15 (GDF-15, a novel targetable immune checkpoint [35]).

VEGF (vascular endothelial growth factor), and other angiopoietic factors, with CXCL5 [36], M-CSF [37], and CD163-expressing MΦs promote angiogenesis [38]. The hepatocyte growth factor (HGF) is a key player in AML cell proliferation [24], drug-resistance, and poor survival [39,40]. The insulin-like growth factor (IGF) and its binding protein insulin-like growth factor binding protein 2 (IGFBP-2), secreted by HL-60 and U937 cells, have important functions in promoting blast proliferation/survival and resistance to chemotherapy [41,42,43]. Cytokines, such as IL-18 and the IL-18-binding protein, gained increasing attention for their roles in MΦ activation syndrome [44]. Furthermore, M-CSF, macrophage migration inhibitory factor (MIF), IL-8, and CCL2 may promote a protumoral MΦ polarization [25,45] contributing to forming a protective niche for leukemia stem cells [17].

CXCL10 could contribute to either a protumoral or an antitumoral activation, depending on the context [46]. The relative levels of cytokine secretion strongly differ between patients and/or leukemia cell lines, which is consistent with the heterogeneity of AML genetics, phenotypes, and biology and may contribute to the variability in the expression of markers associated to HD^AML^-MΦ polarization, such as CD163, CD80, and CD206.

### 3.3. HD Macrophages Promote Primary Myeloblast Survival

Since myeloblasts can polarize monocyte-derived MΦs (Figure 1C), we examined whether co-culturing primary human myeloblasts with HD monocytes (HD Mo) would affect blast cell survival and proliferation. Myeloblasts obtained from nine newly-diagnosed patients were cultured alone or with freshly isolated HD Mo, in direct contact or separated by a TW, for 7 days. Myeloblasts labeled with PKH26 did not reveal significant changes in proliferation when they interacted with HD Mo (T. S., personal observation, 2020). However, the co-culture of six patient cells in direct contact with HD Mo significantly improved their survival at day 7, compared to culture in plastic (Figure 2A).

This survival advantage was dependent on direct contact with the MΦs, as it was abrogated when blasts and MΦs were separated by TW inserts. These observations suggest that the survival of certain sub-types of AML could be dependent on direct interactions with MΦs. In contrast, the co-culture of normal HSPC, under the same conditions, with HD Mo did not affect their survival (Appendix A).

The cross-talk between myeloblasts, MΦs, and other components of the BM microenvironment induces the secretion of multiple proinflammatory and protumoral growth factors, which promote the remodeling of the leukemia stem cell niche and may stimulate leukemia cell proliferation and/or survival [47]. The identification of the involved soluble molecules (SM) and/or their receptors is important as they may be therapeutic targets. SM resulting from the interplay between HD Mo co-cultured with primary myeloblasts from patients #3 and #23 were identified after 7 days using a cytokine array (Appendix A, Appendix A).

The results were compared to those of myeloblasts cultured without monocytes (Appendix A) and both conditions were illustrated in Figure 2B. Common and distinct characteristics are shared by AML#3 with favorable vs. #23 with adverse genetic risk. Several survival and proliferation factors were clearly upregulated when myeloblasts were in contact with HD Mo as opposed to cultured on plastic, such as IGFBP-2, IL-10, and epidermal growth factor (EGF) [48] (Figure 2B).

HGF was secreted at higher levels in co-culture with AML#23. M-CSF was upregulated in co-culture with AML#3, involved in paracrine cross-talks previously observed to drive solid tumor invasion [32,49]. Adhesion receptors VCAM-1 (vascular cell adhesion protein 1) and ICAM-1 (intercellular adhesion molecule-1) were both increased in the co-culture medium from patient#23. Several proangiogenic factors were strongly upregulated by co-culture, including angiogenin, PDGF-AA (platelet-derived growth factor), -AB/BB, and fibroblast growth factor 19 (FGF-19). Moreover, CXCL10, CXCL12, and CCL7, which may promote tumor progression, were also highly induced [32,50,51]. IL-17A, known to promote multiple myeloma cell proliferation, was highly secreted by primary myeloblasts co-cultured with MΦs [52].

### 3.4. CD163^+^ MΦs Activated with M-CSF Support Myeloblast Survival and Proliferation

As MΦs can contribute to creating a protumoral blast niche in BM microenvironment, we tested more specifically the effect of MΦ orientation on blast survival and proliferation. Myeloblasts were co-cultured for 4 days with HD MΦs, which had been oriented during one week in medium containing GM-CSF (GM-MΦs) or M-CSF (M-MΦs). Co-cultures were performed in PM, either in direct contact or separated with a TW.

The analysis of PKH26 dilution at day 4 showed that myeloblasts co-cultured with the CD163^−^ GM-MΦs exhibited a significantly lower proliferative activity, correlated to higher PKH26 fluorescence, compared to those cultured on CD163^+^ M-MΦs (Figure 2C). Moreover, a significantly higher proportion of AnnV^+^ myeloblasts was detected, when they were cultured in contact with GM- than with M-MΦs (Figure 2D). Interestingly, in the absence of direct contact between myeloblasts and GM-MΦs, blasts in TW inserts largely escaped from apoptosis. By contrast, the proportion of AnnV^+^ blasts co-cultured with M-MΦs did not significantly differ in the presence or absence of contact.

### 3.5. CSF1R Inhibition and Exposure to GM-CSF Reprograms CD163^+^ M- and HD^AML^-MΦs

As the upregulation of CD163 expression on HD^AML^-MΦs can be prevented by the inhibition of CSF1R activity (Figure 1D), we assessed whether selective CSF1R tyrosine kinase inhibitors GW2580 [25] or PLX3397 [53] could modify MΦ polarization. As illustrated in Figure 3A, high CD163 expression on M-MΦs was maintained after 7 additional days of culture in M-CSF (week#2, Figure 3A). CSF1R inhibition or addition of GM-CSF during week#2 lowered CD163 expression, with a stronger effect when GM-CSF was used in combination with either CSF1R inhibitor. In parallel, all treatments also significantly increased CD80 surface expression (Appendix A).

The effect of M-MΦ reprogramming on myeloblast proliferation was assessed by comparing the proliferation of HL-60, MV-4-11, NB4, U937, and OCI-AML3 myeloblasts co-cultured for 4 days with M- vs. R-MΦs. Compared to myeloblasts in direct contact with CD163^+^ M-MΦs, the co-culture with R-MΦs (reprogrammed with GM-CSF combined with either GW2580 = R^GM/GW^, or PLX3397 = R^GM/PLX^), significantly slowed down their proliferation and promoted myeloblast apoptosis (Figure 3B,C). Interestingly, a significant decrease in myeloblast apoptosis was observed when they were separated from R-MΦs by TW inserts. We next assessed the survival of primary myeloblasts of all three risk categories cultured on R-MΦs (Figure 3D): their proliferation was not significantly affected by R-MΦs (T. S., personal observation, 2020), but their survival was strongly and significantly reduced compared to survival on M-MΦs.

We then investigated the impact of HD^AML^-MΦ reprogramming: after 7 days of reorientation, single treatments significantly lowered CD163 expression, whereas the combination of GW2580 or PLX3397 with GM-CSF almost abolished it (Figure 4A); in parallel, reorientation increased CD80 expression (Appendix A). The co-culture of HL-60, NB4, and U937 cells with HD^AML^ R-MΦs, lowered their proliferative activity (Figure 4B) and significantly induced their apoptosis (Figure 4C). As observed with R-MΦs (Figure 3C), the separation of myeloblasts from HD^AML^ R-MΦs by a TW, protected them from apoptosis (Figure 4C). These findings reveal a major role for HD^AML^-MΦ polarization, and most likely for adhesive interactions, in controlling myeloblast survival.

The impact of MΦ reprogramming using CSF1R inhibitors combined to GM-CSF was then assessed on autologous primary BM MΦs from AML patients. BM cells were plated at diagnosis, and MΦs were co-cultured with myeloblasts, leukocytes, and stroma for 7 days under the conditions indicated in Figure 4D. Autologous BM AML-MΦs maintained their CD163 positivity in PM, as in medium supplemented with M-CSF.

CSF1R inhibition with GW2580 was least efficient, while GM-CSF supplementation reduced CD163 expression, but not as strongly as when combined with GW2580. The combination of PLX3397 with GM-CSF also efficiently inhibited CD163 surface expression. In parallel, we monitored CD80 expression, which significantly increased with GM-CSF either alone or combined with GW2580 or PLX3397 (Appendix A). Finally, we measured primary patient blast apoptosis in four different BM patient co-cultures in medium with M-CSF vs. reprogramming conditions; twice as many apoptotic blast cells were found in both reprogramming media compared to what was observed in M-CSF (Appendix A).

Innate and adaptive immune checkpoint deregulation have been shown to play important roles in myeloid malignancies. We, therefore, analyzed the expression of CD47, which couples to SIRPα on MΦs [14,54], and of PD-L1 [15]. Interestingly, PD-L1 expression was unchanged or even increased on CD163^low^ HD or autologous primary MΦs exposed to CSF1R inhibitors, compared to CD163^high^ cells cultured with M-CSF (Appendix A). Further FC analyses showed that the reprogramming of autologous MΦs co-cultured with primary myeloblasts did not change CD47 on myeloblasts nor SIRPα expression on MΦs (Appendix A).

Taken together, our results indicate that CSF1R plays a critical role in the polarization of HD M-MΦs and of allogeneic HD^AML^- or autologous AML-MΦs toward a protumoral M2-like phenotype and that inhibiting the CSF1R activity, in the presence of GM-CSF, efficiently reverses MΦ orientation. This is evidenced by a down-regulation of CD163 and an increase of CD80 expression, which has a direct impact on myeloblast proliferation and survival.

### 3.6. Macrophage Reprogramming Reverses Drug Resistance

We next assessed whether MΦ orientation affects the sensitivity of myeloblasts to targeted therapy. We first examined whether MΦ polarization has an effect on myeloblast sensitivity to venetoclax, a BCL-2 inhibitor. Myeloblasts were cultured in direct contact with HD M- vs. GM-MΦs (week#1) or M- vs. R-MΦs (week#2) in presence of vehicle or escalating doses of venetoclax. After 48 h, the percentage of live cells (AnnV^−^ and 7-AAD^−^) was measured by FC.

We observed that the co-culture of NB4 cells with M-MΦs induced blast resistance to venetoclax, as compared to blasts cultured alone (T. S., personal observation, 2020). Direct contact contributed to the resistance of myeloblasts to venetoclax, as NB4 cells became significantly more sensitive to it, when they were separated from M-MΦs by TW inserts (Figure 5A). By contrast, venetoclax strongly induced NB4 cell apoptosis, even at low concentrations, when blasts were cultured on GM-MΦs (Figure 5A). Importantly, this was also the case when NB4 were cultured on R-MΦs (reprogrammed with either GW2580 or PLX3397, and GM-CSF, R^GM/GW^-, or R^GM/PLX^-MΦs; Figure 5B). MΦ reprogramming with GW2580 and GM-CSF also significantly sensitized OCI-AML-3 myeloblasts to venetoclax (Figure 5C).

FLT3-mutated AML have a poor prognosis [55]. To determine the impact of MΦ orientation and reprogramming on myeloblasts sensitivity to midostaurin (a FLT3 inhibitor used in AML patients [56]), we co-cultured the MLL-AF4 and FLT3-ITD mutated MV-4-11 AML cells with M- vs. GM- (week#1) or M- vs. R-MΦs (week#2). MV-4-11 cells co-cultured in direct contact with M-MΦs were resistant to midostaurin (Figure 5D). In contrast, when myeloblasts were separated from M-MΦs by a TW insert, midostaurin efficiently induced MV-4-11 apoptosis, suggesting an important role for cell adhesion in promoting resistance to midostaurin (Figure 5D). As observed with venetoclax, the co-culture of MV-4-11 myeloblasts on GM- or R-MΦs strongly increased their sensitivity to midostaurin (Figure 5D,E). Taken together, our results suggest a critical role of MΦ orientation in promoting drug-resistance to midostaurin and venetoclax.

### 3.7. Factors Secreted by GM- and R-MΦs Play a Role in Myeloblast Apoptosis

In order to assess whether factors secreted by MΦs could also have an impact on myeloblasts, HL-60, NB4, OCI-AML3, and U937 cells were cultured with MΦ CM. Of the four tested cell lines, only U937 displayed sensitivity to MΦ CM: after 24 h, GM-MΦ CM significantly impaired their proliferation, while R^GM/GW^-MΦ CM strongly decreased it (Figure 6A). Moreover, both GM- and R^GM/GW^-MΦ CM induced >25% apoptosis in U937 cells (Figure 6B).

Effects on proliferation and apoptosis were already detectable at 6 h (C. S., personal observation, 2020). Moreover, after 48 h of incubation with MΦ CM, cleaved caspase-3 was detected by western blot (Figure 6C; uncropped version Appendix A); of note, caspase-3 cleavage was already detectable at 24 h, but less markedly. M-MΦ CM had no impact on apoptosis induction and caspase-3 cleavage. These data indicate at least a partial contribution to apoptosis of both inflammatory/cytotoxic molecules contained in polarized GM- and R^GM/GW^-MΦ CM.

### 3.8. Identification of Soluble Molecules Secreted by MΦs

To facilitate the identification of SM predominantly secreted by M-, GM-, and R^GM/GW^-MΦs, we illustrated several relevant cytokines as their fold-decrease (Figure 7A) or -increase (Figure 7B) in GM- or R-MΦ CM relative to their quantification in M-MΦ CM, while the short tables indicate the SM relative levels (data from all analyzed SM are shown in Appendix A and revealed membranes are shown in Appendix A). M-MΦs secrete prosurvival factors, such as HGF, and EGF, and high levels of VCAM-1, which has been demonstrated to play an essential role in the BM stromal compartment in mediating leukemia cell chemoresistance [57].

Other M-MΦ SM detected in the array may play a role, such as CXCL12 or IL-8 (Figure 7A and Appendix A), as they contribute to support myeloblast survival and proliferation [58,59] and to orient MΦs toward immunosuppressive and proangiogenic functions [60]. Furthermore, M-MΦ reprogramming with GM-CSF and GW2580 reduced the secretion of many factors responsible for survival, angiogenesis, and cross-talk with stromal cells, for example HGF, angiogenin, IGFBP-2, IL-10, and PDGF-AB/BB. The tumor-promoting chemokine CCL7 [61] is also more abundantly secreted by M-MΦs than GM- or R-MΦs. CXCL9 and CXCL10, which may exhibit pro- or anti-tumoral activities, depending on the splice variant of their receptor CXCR3 [62], were reduced in CM of both GM- and R-MΦs [63].

The inflammatory cytokine TNF-α is highly secreted in GM- and R^GM/GW^-MΦ CM (Figure 7B). Other factors, with context-dependent antitumoral or immune cell chemotactic activity, such as CCL5 [64], CCL17 [65], CCL3/4, CCL20 [66], CXCL1, and CCL19 [67], were also predominantly detected in CM of GM- and R^GM/GW^-MΦs. Compared to MΦs stimulated with M-CSF, GM- and R-MΦs secreted increased levels of IL-6 and IL-27, which may also contribute to inhibiting myeloblast proliferation and promoting blast apoptosis [39,68].

We next determined the effect of MΦ reprogramming on soluble factor secretion resulting from the interplay between HD MΦs co-cultured with primary myeloblasts from patient #3 and #23. HD Mo were first stimulated for one week with CM from patient #3 or #23. On day 7, after confirming the induction of CD163 expression in HD^AML^-MΦs, primary blasts of patients #3 or #23 were added to the respective MΦ monolayers for one more week (week#2), in PM vs. PM supplemented with GM-CSF and GW2580 (Appendix A). The level of each SM was illustrated as the ratio of its level in PM supplemented with GM-CSF and GW2580 divided by that secreted by co-cultures in PM (key cytokines in Figure 8).

Consistent with analyses of GM- and R-MΦ CM shown in Figure 7, the relative levels of TNF-α increased in both co-cultures of AML #3 and #23 with HD MΦs in the presence of GM-CSF and GW2580. The secretion of interferon γ, a key inflammatory molecule, also increased in both co-cultures under reprogramming conditions, as were CCL5, CCL17, CCL3/4, CCL19 and CCL20. IL-6 was highly secreted in CM of AML #3 co-cultured with HD^AML^-MΦs in reprogramming conditions. IL-27 exhibited an opposite trend between patients. Interestingly, a number of other SM known for their protumoral activity, such as IL-10, EGF, HGF, IGFBP-2, PDGF-AB/BB, CXCL12, FGF-19, IL-18 Bpa, and VCAM-1, slightly increased or did not exhibit any significant relative change in their levels in this experiment.

### 3.9. TNF-α Induces Apoptosis in U937 Cells

TNF-α is highly increased in GM- and R-MΦ CM (Figure 7B); while it could have a major function in modulating the surrounding microenvironment cross-talks, and inducing the expression and secretion of other inflammatory factors, it was shown to be cytotoxic to U937 cells [69]. We, therefore, tested the effect of escalating doses of recombinant TNF-α on U937 cells. After 24 h, we confirmed that TNF-α triggered apoptosis of U937 cells in a dose-dependent manner (Figure 9A). We next investigated whether adding a neutralizing antibody against TNF-α in MΦ CM would affect the induction of apoptosis. As illustrated on Figure 9B, adding a monoclonal antibody against TNF-α in the CM strongly decreased apoptosis, whereas adding an antibody against IL-6, an inflammatory factor which is also increased in both MΦ CM, had no impact.

## 4. Discussion

AML form a heterogeneous group of diseases, frequently of poor prognosis, with a high risk of relapse after completion of therapy. In patients with solid tumors and leukemia, the presence of TAMs has been reported as a bad prognosis factor, which may be related to the change in MΦ polarization inside tissues infiltrated by malignant cells, promoting tumor growth and survival [23]. As little information is available on AML-associated MΦs, we examined their impact on myeloblast survival, proliferation, and resistance to venetoclax or midostaurin.

In addition, we examined the possibility of changing the MΦ orientation by inhibiting the CSF1R. Our results show that (1) BM AML MΦs frequently express the protumoral orientation marker CD163 at diagnosis; (2) myeloblasts can educate HD MΦs to express CD163 in a CSF1R-dependent manner and promote their growth and survival; (3) numerous SM are secreted that support myeloblasts proliferation, survival, migration, and angiogenesis; (4) resistance to venetoclax and midostaurin is promoted by myeloblast interactions with MΦs; and (5) MΦ orientation can be reprogrammed by inhibition of CSF1R combined with GM-CSF exposure, leading to the reversal of resistance to targeted therapies and myeloblast apoptosis.

MΦs are highly plastic cells that can change their phenotype and function in response to microenvironmental stimuli, which modulate their polarization [70]. In vitro, they adopt either an antimicrobial and antitumoral activity when they are exposed to cytokines secreted by Th1 lymphocytes (such as TNF-α or interferon γ) or an anti-inflammatory phenotype promoting angiogenesis and tumor growth when they are exposed to Th2 lymphocytic cytokines (such as IL-4, -5, and -13) [71]. In tumors, depending on local cytokines secreted by malignant cells and tissue environment, TAM orientation can be shared between M1 and M2 status, with multiple phenotypes between these two extremes [15].

In the majority of AML patients, we observed that the BM and PB MΦs significantly express the M2-like marker CD163 at diagnosis (Figure 1A). Interestingly, similar observations were reported in B- and T-cell acute lymphoblastic leukemia, suggesting that leukemia cells and their microenvironment may dysregulate MΦ function [72]. In a minority of AML patients, we also observed MΦs expressing CD80, suggesting that a continuum of phenotypes between M1- and M2-like extremes are present at diagnosis, whose function may change in response to local signals.

The co-culture of AML cell lines or primary myeloblasts with HD Mo highly induced CD163 expression on HD^AML^-MΦs (Figure 1C), mimicking the effect of M-CSF (Figure 1B). This induction did not require direct contact between myeloblasts and MΦs as it was also observed with exposure to AML CM, suggesting that it may depend on soluble factors, mainly cytokines [73]. However, cytokines known to induce the expression of CD163, such as IL-4, -13, and -10 [74], were not detected in AML CM (Appendix A). Alternatively, we hypothesized that it may depend on CSF1/CSF1R pathway activation, as in solid cancers [15] and lymphoma [27,28], which was confirmed by the abrogation of CD163 expression induced by AML CM supplemented with GW2580 or PLX3397 (Figure 1D).

Previous observations reported a sensitivity of primary AML cells to CSF1R inhibition, in favorable-risk patients, through the secretion of cytokines, in particular HGF, by CSF1R-expressing supportive cells [24]. Inhibiting CSF1R in order to reprogram protumoral MΦ was crucial in our experiment, but MΦs were simultaneously cultured with GM-CSF to induce an M1-like polarization. GM-CSF, as a key driver of inflammatory reaction, contributes to the antitumoral effect of R-MΦs. It may do so by enhancing the MΦ phagocytic activity, the production of reactive oxygen species, and the secretion of proinflammatory cytokines that affect T-lymphocyte response against malignant cells [75,76].

M-MΦs (Figure 3A), HD^AML^-MΦs (Figure 4A), and primary autologous MΦs in their BM microenvironment (Figure 4D) were efficiently reprogrammed by CSF1R inhibition leading to a drastic decrease in CD163 expression; addition of GM-CSF potentiated this down-regulation. In co-culture assays, R-MΦs efficiently induced primary myeloblasts and AML cell line apoptosis, but as with GM-MΦs, myeloblasts were partially rescued from apoptosis in the absence of direct contact, suggesting a role for cell adhesion molecules. However, SM also participated in the proapoptotic effect of R-MΦs. CM from GM- or R-MΦs induced caspase-3 cleavage and the apoptosis of >25% of U937 monoblasts.

Interestingly, mAb blocking studies indicated that TNF-α may contribute to inducing U937 apoptosis. High TNF-α levels were detected in CM of GM- and R-MΦs (Figure 7B) and recombinant TNF-α induced U937 cell apoptosis. In myeloblasts, TNF-α could activate the apoptosis cascades through binding to its receptor TNFR1, FADD (Fas-associated protein with death domain) recruitment, and caspase-8 and -3 activation or by activation of the mitochondrial pathway leading to caspase-9 and -3 activation [77].

In a mouse model of chronic lymphocytic leukemia, targeting MΦs using an anti-CSF1R mAb efficiently inhibited disease progression; interestingly, the leukemic cell death was dependent on TNF-α signaling and tumor microenvironment reprogramming toward an antitumoral phenotype [78]. In vivo, TNF-α could have not only an effect on myeloblasts but also a major impact on MΦ orientation by counterbalancing the emergence of M2-like AML-MΦs by inhibiting IL-13, and possibly IL-4, secreted by eosinophils in tumors [79].

Moreover, TNF-α can induce the secretion of other proinflammatory effectors, like GM-CSF, which activates interferon regulatory factor 4 (IRF4) and induces the biosynthesis of CCL17 in MΦs, a cytokine loop with proinflammatory properties [80] promoting MΦ M1-like orientation and myeloblasts apoptosis. However, the CM of GM- or R-MΦs had no proapoptotic effect on three other AML cell lines; this may be due to the ability of TNF-α to simultaneously activate pathways leading to cell apoptosis or survival and proliferation [77].

Other mechanisms may contribute in vivo to the proapoptotic effect of R-MΦs. Figure 7 shows that the cytokine profile of R-MΦs differs from that of GM-MΦs by the secretion of higher levels of chemokines, such as CCL20, CCL19, IL-6, and IL-27. The crucial role played by CCL19 in the immune response is illustrated by its ability with IL-7 to promote tumor infiltration by T cells and dendritic cells and to improve the therapeutic effect of CAR-T cells against solid tumors [81] or multiple myeloma [82].

The proinflammatory molecule IL-6 may also contribute to immune response by promoting T-cell-mediated immune defense [83] and was even reported to stimulate myeloblast differentiation by MΦs [84]. Finally, blocking CSF1R in vivo inhibits monocyte recruitment into the tumor environment, leading to MΦ depletion. MΦ reprogramming induced by CSF1R inhibition may also strengthen immune defense by stimulating T-cell recruitment in a malignant cell environment [85].

Drug resistance is a major problem in relapsed/refractory AML. The combination of the BCL-2 inhibitor venetoclax with hypomethylating agents or low dose cytarabine demonstrated high initial efficiency in patients with de novo AML, who were ineligible for intensive chemotherapy [86]. However, the response to treatment is most often lost after several months, with poor survival of relapsing patients [87].

Several mechanisms of resistance to venetoclax have been reported [88] involving the up-regulation of antiapoptotic molecules Mcl-1 and Bcl-xL, mutations in genes controlling different kinase pathways, transcription factors, epigenetic modifiers, and tumor suppressors. Similarly, drug resistance affects the sensitivity of FLT3-mutated AML to midostaurin. Like for venetoclax, the mechanisms of resistance may be complex and involve the intrinsic properties of AML cells as well as of the microenvironment [89].

The ability of reprogrammed macrophages to increase the sensitivity of myeloblasts to midostaurin or venetoclax suggests a role for macrophage polarization in promoting resistance to these targeted therapies. Interestingly, close contacts between macrophages and myeloblasts were required to promote resistance to midostaurin or venetoclax, which may suggest the involvement of adhesion receptors. The presence of high soluble VCAM-1 levels in M-MΦ CM and its strong decrease in the CM of GM- and R-MΦs may suggest the involvement of VCAM-1/CD49d, which were previously reported to trigger NF-κB signaling and promote chemoresistance [57].

Aiming to modulate the MΦ phenotype in AML is a promising therapeutic approach. Currently, clinical development of CSF1R inhibitors, including PLX3397, for AML treatment is at the early stages [90]. The efficacy of PLX3397, a FLT3 and CSF1R inhibitor, was demonstrated in relapsed/refractory FLT3-ITD-mutated AML with a safety profile similar to that of other FLT3 inhibitors [91]. Its impact on CSF1R inhibition is unclear in relapsing FLT3-ITD AML, and its clinical activity is predominantly related to its ability to inhibit FLT3 [91]. CSF1R inhibition alone may not be sufficient to overcome resistance mechanisms in vivo [92,93], but it may represent a general approach to target the microenvironment of AML cells and may be promising when used in combination with inhibitors of the immune checkpoints, angiogenesis, or with the adoptive T-cell transfer, which are undergoing clinical investigations [94].

To reprogram human MΦs in AML, combining CSF1R inhibition with GM-CSF might be more efficient for inducing an antitumoral response. It may however not be sufficient to restore MΦ phagocytic activity and antigen-presenting capacity in vivo. Indeed, PD-L1 expression is unchanged or higher on CD163^−^ MΦs exposed to CSF1R inhibitors, than on CD163^+^ cells. PD1/PD-L1 inhibition may improve the antitumoral activity of R-MΦs and conversely, CSF1R inhibition may improve the effect of anti-PD1/PD-L1 monoclonal antibody in relapsed/refractory AML.

In addition to the T-cell immune checkpoints, CD47 is the dominant MΦ checkpoint, which is overexpressed in myeloid malignancies and inhibits phagocytosis through “do not eat me” signals upon its binding to SIRPα on MΦs [54]. As observed for PD1/PD-L1, the reprogramming of AML-MΦs co-cultured with primary myeloblasts did not change CD47 expression on myeloblasts nor SIRPα expression on macrophages. As the combination of the anti-CD47mAb, magrolimab, with 5-azacitidine strongly improves the antitumoral activity of MΦs [54], considering a combination of CD47 blockade and CSF1R inhibitors, with 5-azacitidine, might be a way to improve the in vivo antitumoral activity of reprogrammed MΦs.

## 5. Conclusions

Relapse and resistance to treatment remain major issues in the treatment of AML. We showed here that protumoral CD163^+^ MΦs predominate, at diagnosis, in the BM of AML patients and that the induction of CD163 expression on MΦs by myeloblasts was dependent on CSF1R. We observed that numerous SM were secreted by MΦs and myeloblasts, which promoted an M2-like MΦ orientation, leading to increased blast survival and drug resistance in the presence of close intercellular contacts. Finally, inhibiting CSF1R, in the presence of GM-CSF, reprogramed MΦ orientation, reversed the resistance to targeted therapies, and promoted myeloblast apoptosis. CSF1R inhibition may serve as a novel way to target the microenvironment of myeloblasts and improve the efficiency of AML therapy.

## Figures and Tables

**Figure 1 cancers-13-05289-f001:**
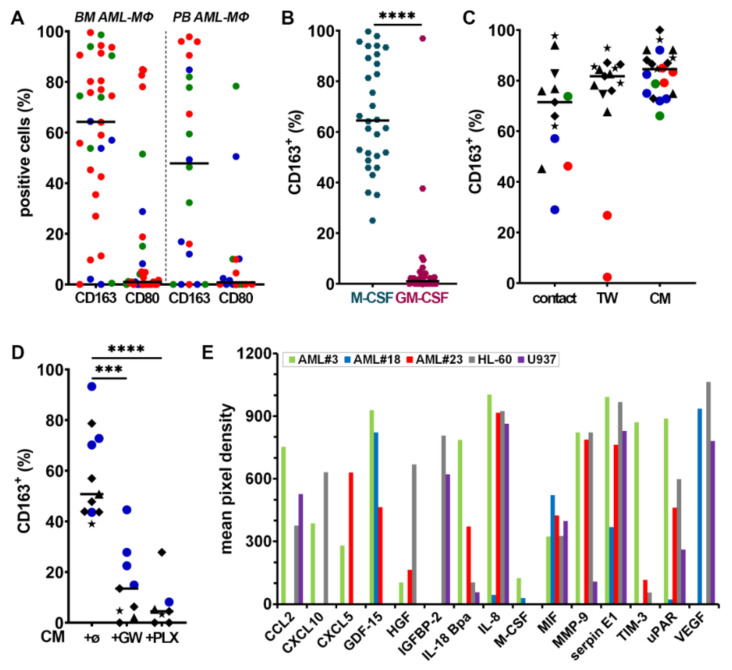
CD163 expression on MΦs is modulated by stimuli and depends on CSF1R activity (**A**) Expression frequency of CD163 and CD80 in BM and/or PB of 42 AML patient MΦs (within CD14^+^/CD45^+^ gate) at diagnosis. Each patient is color-coded according to ELN genetic risk (green = favorable, blue = intermediate, and red = high risk). The median is indicated by a black horizontal line. (**B**) Expression frequency of CD163 in HD MΦs after one week of culture in PM supplemented with M- or GM-CSF, *n* = 30. (**C**) CD163 expression allogeneic HD^AML^-MΦs obtained from monocytes cultured with AML cell lines (★ = HL-60, ▲ = NB4, ♦ = U937, and ▼ = OCI-AML3) or primary patient blasts (dots color-coded according to genetic risk) either in direct contact or separated with a TW. AML conditioned medium (CM) was also tested. The median is indicated by a black horizontal line, *n* = 13–21. (**D**) HD Mo were cultured for 7 days with primary patient blasts (blue) or leukemia cell lines (symbols) CM, supplemented with GW2580 (GW) or PLX3397 (PLX) before analyzing CD163 expression by FC. CM from: ★ = HL-60, ▲ = NB4, and ♦ = U937, blue dot = patient#18; *n* = 3–6 HD. (**E**) Relative quantification of selected cytokines secreted by patient blasts with three different genetic risks and two leukemia cell lines. *** *p* < 0.001 and **** *p* < 0.0001 (M-W test).

**Figure 2 cancers-13-05289-f002:**
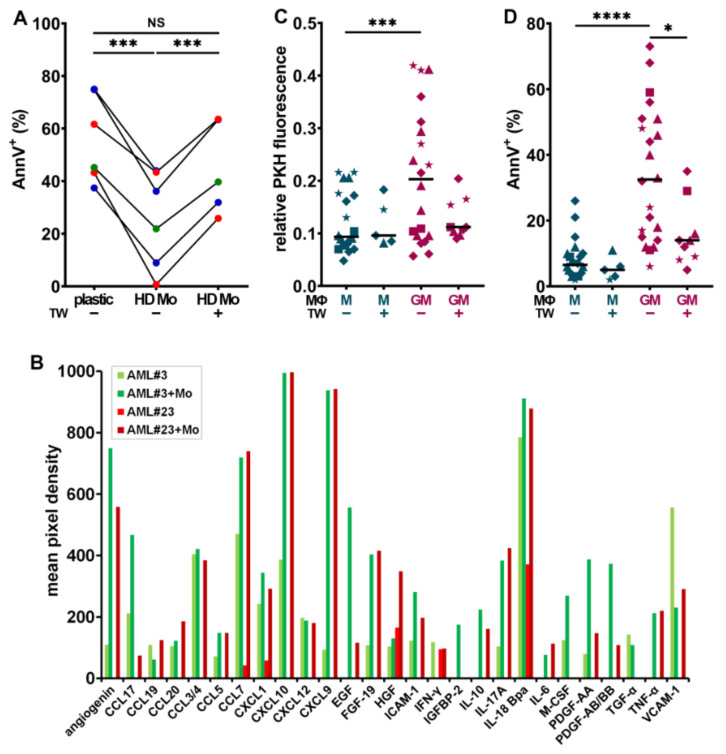
CD163^+^ MΦs support myeloblast survival and proliferation. (**A**) Frequency of AnnV^+^ AML blasts after 7 days of culture alone (plastic), or with monocytes either in direct contact (HD Mo) or separated by a TW. *n* = 6 patients (#9, 11, 17, 18, 26, and 42), color-coded according to ELN genetic risk. *** *p* < 0.001 (paired *t*-test). (**B**) Selected cytokines that were increased in CM when primary myeloblasts (favorable, green vs. high risk, red) were cultivated for 7 days with HD Mo vs. alone are illustrated. (**C**,**D**) PKH-labeled leukemia cell lines were cultured for 4 days on a monolayer of M- or GM-MΦs and analyzed for proliferation (**C**) and survival (**D**). * *p* < 0.05 (M-W test), *** *p* < 0.001, and **** *p* < 0.0001 (WMP test). (★ = HL-60, ■ = MV-4-11, ▲ = NB4, and ♦ = U937).

**Figure 3 cancers-13-05289-f003:**
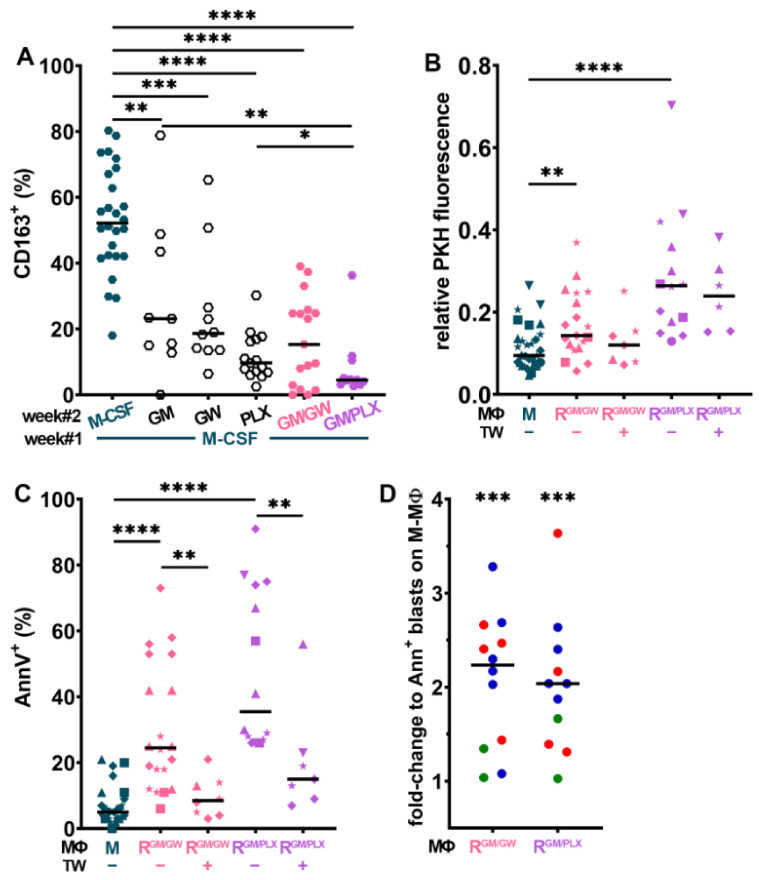
R-MΦs induce myeloblast apoptosis. (**A**) Expression frequency of CD163^+^ MΦs whose culture medium has been supplemented with M-CSF for 2 weeks or switched to reorienting medium with GM-CSF (GM) and/or GW2580 (GW) and/or PLX3397 (PLX) for one week, *n* = 11–26. (**B**,**C**) PKH-labeled leukemia cell lines were cultured for 4 days on monolayers of M- or R^GM/GW^- or R^GM/PLX^-MΦs +/− TW and analyzed for proliferation (**B**) and survival (**C**), *n* = 14–30. ★ = HL-60, ■ = MV-4-11, ▲ = NB4, ♦ = U937, and ▼ = OCI-AML3. (**D**) Primary patient myeloblasts color-coded according to ELN genetic risk were co-cultured for 4 days on monolayers of M- or R^GM/GW^- or R^GM/PLX^-MΦs and analyzed for survival; AnnV positivity is expressed as fold-change compared to culture on M-MΦs. *n* = 11–12. * *p* < 0.05, ** *p* < 0.01, *** *p* < 0.001, and **** *p* < 0.0001 (using M-W test (**A**–**C**) or WMP test (**D**)).

**Figure 4 cancers-13-05289-f004:**
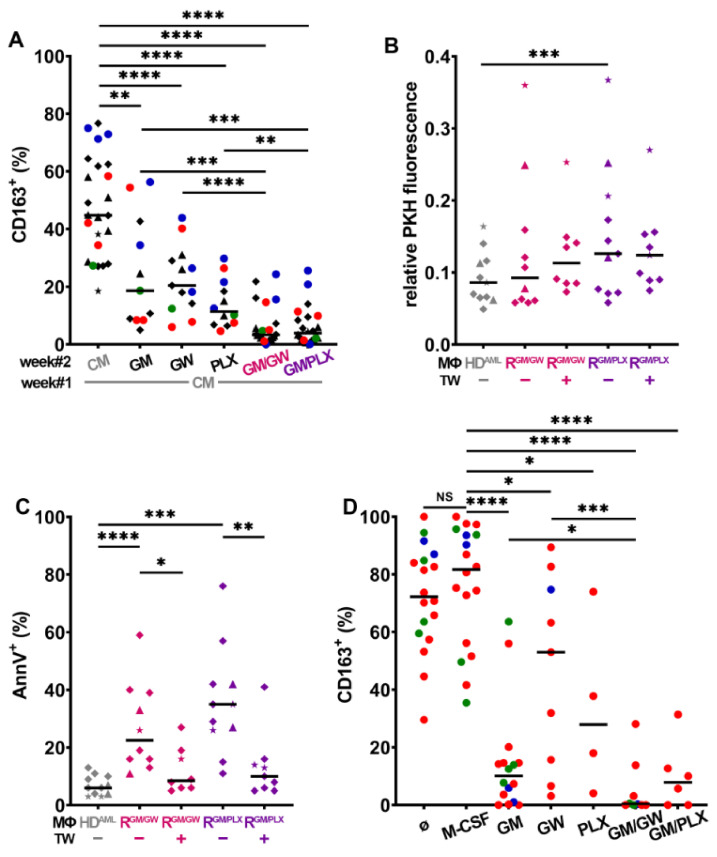
Reprogrammed AML macrophages induce myeloblast apoptosis. (**A**) After one week of stimulation in CM from U937 (♦), HL-60 (★), NB4 (▲), or primary patient blasts (colored dots), HD^AML^-MΦs were either left in the same CM or reoriented in PM with indicated supplements for one additional week and CD163 expression was analyzed by FC. *n* = 11–24. (**B**) Myeloblast proliferation and (**C**) apoptosis induced after 4 days of co-culture on HD^AML^-MΦs. ★ = HL-60, ▲ = NB4, and ♦ = U937. (**D**) Primary BM patient samples color-coded according to genetic risk were cultured for 7 days in PM + indicated supplements and analyzed by FC. (GM/GW = GM-CSF + GW2580, GM/PLX = GM-CSF + PLX3397). * *p* < 0.05, ** *p* < 0.01, *** *p* < 0.001, and **** *p* < 0.0001 (using M-W test except between paired groups that were compared with WMP test).

**Figure 5 cancers-13-05289-f005:**
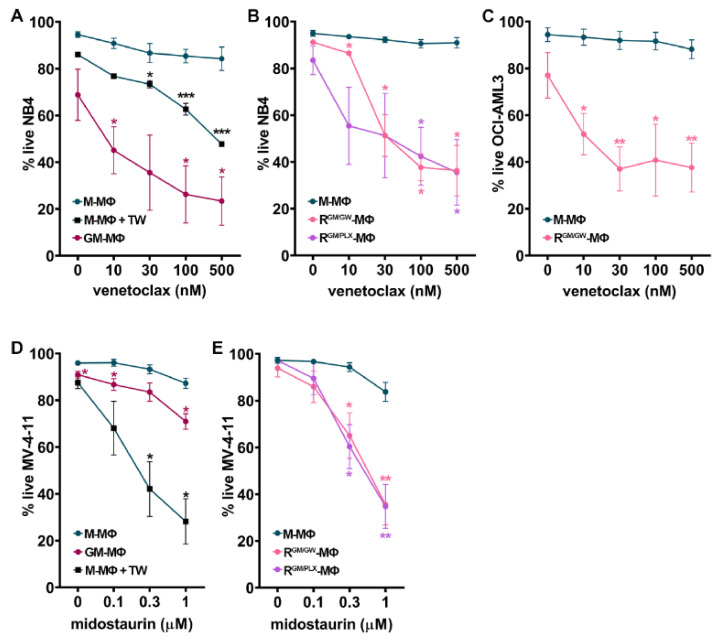
MΦ reprogramming reverses myeloblast resistance to targeted therapy. AML cell lines were cultured with indicated MΦ monolayers for 48 h, and the frequency of live (AnnV/7-AAD-negative) cells was measured by FC in the presence of escalating doses of venetoclax (**A**–**C**), or midostaurin (**D**–**E**). (**A**) Frequency of live NB4 cells after culture with GM- or M-MΦs either in direct contact or separated by a TW. (**B**) Frequency of live NB4 cells cultured with MΦ whose culture medium has been supplemented with M-CSF for 2 weeks (M-MΦs) or switched to reorienting medium supplemented with GM-CSF (GM) and GW2580 (GW) or PLX3397 (PLX) (R^GM/GW^- or R^GM/PLX^-MΦs) during week#2. (**C**) Frequency of live OCI-AML3 as above (**B**) cultured with M- or R^GM/GW^-MΦs. (**D**) Frequency of live (7AAD-negative) MV-4-11 cells after culture with GM- or M-MΦs either in direct contact or separated by a TW. (**E**) The % of live MV-4-11 cells cultured with MΦs whose culture medium has been supplemented with M-CSF for 2 weeks (M-MΦs) or switched to reorienting medium (R^GM/GW^- or R^GM/PLX^-MΦs). Experiments were performed using monocytes isolated from 3–13 different HD. Data points represent the mean ± SEM; * *p* < 0.05, ******
*p* < 0.01, and *** *p* < 0.001 to M-MΦs (M-W test).

**Figure 6 cancers-13-05289-f006:**
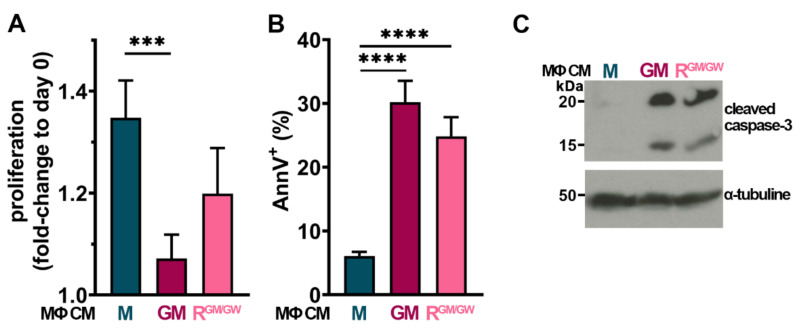
GM- and R- MΦ CM impair U937 proliferation and induce their apoptosis. (**A**,**B**) U937 cells were cultured in PM supplemented with 25% MΦ CM and counted after 24 h, *n* = 5–8 (**A**) or analyzed by FC with AnnV staining, *n* = 7–13 (**B**). The mean + SEM is plotted for each condition. *** *p* < 0.001, and **** *p* < 0.0001 (M-W test). (**C**) Lysates from 10^6^ U937 cells cultivated for 48 h with indicated MΦ CM were immunoblotted with anti-cleaved caspase-3 antibody (upper panel) or anti-α-tubulin (lower panel) antibody as control.

**Figure 7 cancers-13-05289-f007:**
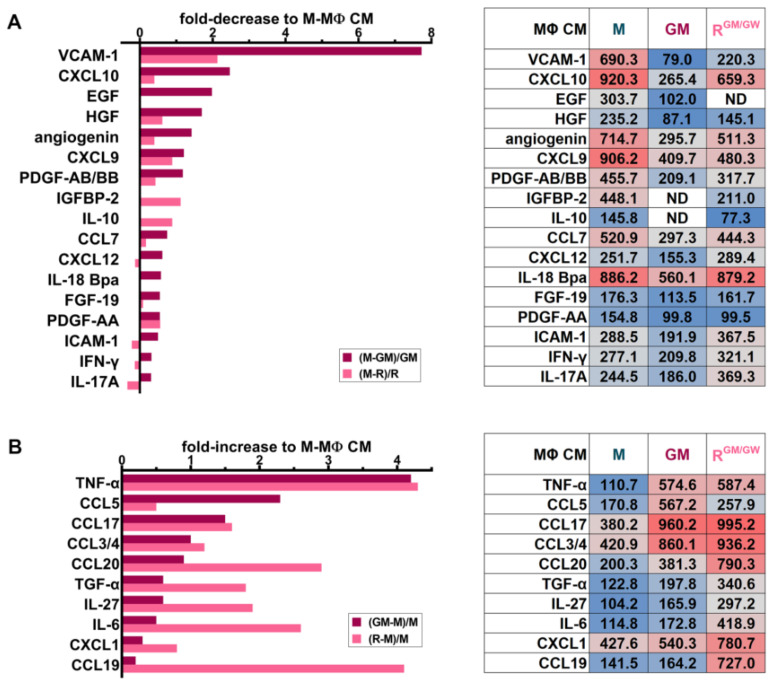
Cytokine secretion profiles differ with MΦ polarization status. HD monocytes were stimulated for 7 days with M- or GM-CSF and CM was collected and analyzed. Alternatively, CM of MΦs whose culture medium has been supplemented with M-CSF for 1 week and switched to reorienting medium supplemented with GM-CSF (GM) and GW2580 (GW) for one more week was also collected. (**A**) A selection of cytokines is illustrated, which are decreased in GM- (dark red) and R-MΦ (pink) CM compared to M-MΦ CM, according to the formula in graph legend. (**B**) Selected increased cytokines in GM- (dark red) or R-MΦ (pink) CM compared to M-MΦ CM are illustrated and calculated according to the formula in the graph legend. For both A and B, the relative optical density of the selected cytokines are indicated in the tables and colored from blue (low) to grey (medium) to red (high) according to their mean pixel intensity.

**Figure 8 cancers-13-05289-f008:**
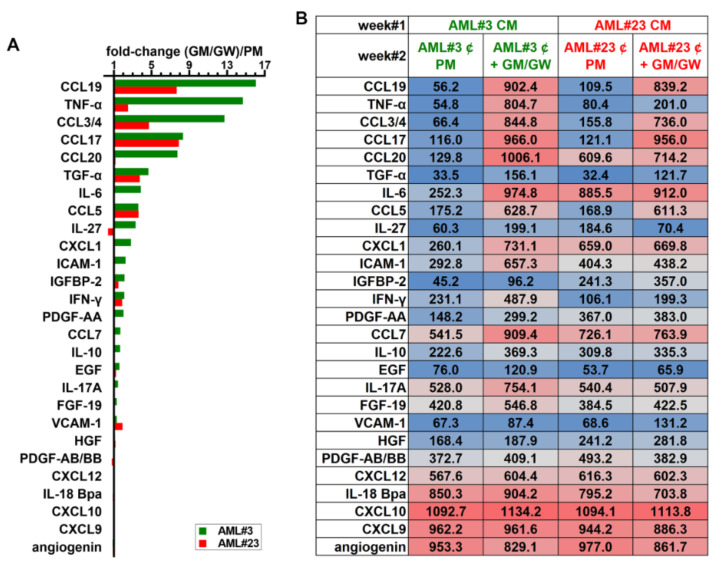
The secretion of proinflammatory cytokines is increased in co-cultures of primary AML blasts with reprogrammed HD^AML^-MΦs. HD Mo were stimulated for 7 days with CM from primary blasts from patients #3 or #23 and primary blasts from the same patients were then added to the respective HD^AML^-MΦs with or without GM-CSF + GW2580 for 7 more days (week#2). At the end of week#2, CM from the four distinct co-cultures were collected and analyzed in cytokine arrays. (**A**) Fold change in relative secretion of selected cytokines secreted by blasts from patient #3 (favorable risk, green bars) and #23 (high risk, red bars) co-cultured with reprogrammed HD^AML^ MΦs in PM containing GM-CSF and GW2580, divided by their respective relative levels in control co-cultures of AML blasts with HD^AML^ MΦs in PM. (**B**) Relative optical density of the selected cytokines illustrated in the graph colored from blue (low) to grey (medium) to red (high) according to their mean pixel density.

**Figure 9 cancers-13-05289-f009:**
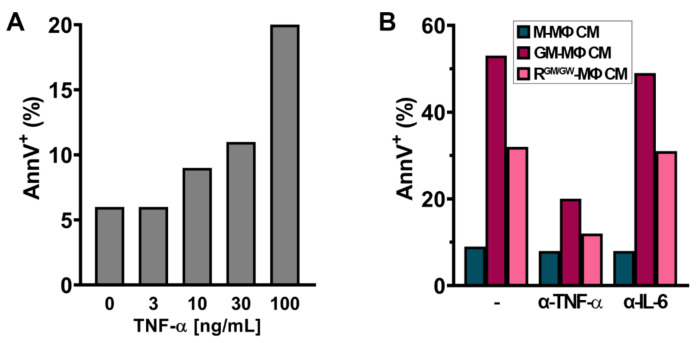
TNF-α in MΦ CM can trigger apoptosis in U937 cells. (**A**) U937 cells were cultured for 24 h with increasing concentrations of TNF-α and apoptosis was monitored by FC. (**B**) U937 cells were cultured for 24 h with 25% indicated MΦ CM and anti-TNF-α (30 µg/mL) or anti-IL-6 (300 ng/mL) antibodies. AnnV positivity was analyzed by FC.

## Data Availability

Data is contained within the article or Appendix A.

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
