# Peer review of "CSF1R Inhibition Combined with GM-CSF Reprograms Macrophages and Disrupts Protumoral Interplays with AML Cells"

_cancers, 2021, doi:10.3390/cancers13215289_

Round 1
Reviewer 1 Report
In the study CSF1R inhibition combined with GM-CSF reprograms macrophages and disrupts antitumoral interplays with AML, Smirnova et al described the interplay between AML cells and tumor associated macrophages.
While the topic is of interest for the field, there are major flow in the study and the conclusions cannot be supported by the results. Therefore the manuscript cannot be accepted in the present form
- The author should add HSPC as control for their experiments with AML cells. Indeed they cannot rule out if the effect on macrophages or of macrophages are specific to AML
- Line 228-232: why so many data not shown?
- The authors should add HDHSPC Macrophages (stimulated with HSPC supernatant) for control of HDAML macrophages
- To conclude on the role of CSF1R, the authors should add either a knock down of CSF1R or show the inhibition of CSF2R and activation of CSF1R
- It is not clear how in their surface marker analysis the author separate macrophages from AML cells in all co-culture assays (including transwell). The authors should not forget that AML cells can migrate through the membrane as macrophages.
- Minor: I would suggest the authors to write their manuscript in a lighter style because it is not always easy to follow the authors perspective
Reviewer 2 Report
The present manuscript from Smirnova et al focusses on the interplay between Acute Myeloid Leukemia blasts and macrophage polarization, mainly through the stimulation by secreted cytokines highlighting the role of GM-CSF and CSF1R in this model to govern the M2 polarization as evidenced using anti-CD163 antibodies.
The manuscript is original and of interest for the readers of Cancers journal but should be modified prior to publication in order to be clearer and better organized.
Major concerns:
- I do not have access to supplementary tables S1-S2-S3 to properly evaluate the results. Please provide them in the revised version within the pdf document with supp figures. The authors used 42 newly diagnosed AML. Are the different subtypes representative of AML? in other words, is the distribution of the AML sub-types similar to the well-known distribution described in the literature?
- In a general manner, the figures are not always easy to look at (Figure 1E, 2B, 7A-B, 8A). For figures 7A-B and 8A, I would recommend to represent the graphs as horizontal bars that could be positioned in front of the corresponding lanes of the relative tables in Figure 7C and 8B. Figure 1A : Y-axis should be % positive cells (not expression). In all plot presentations, the points corresponding to the HL-60, NB4 and U937 cell lines are not easy to look at and should be increased. For figure 1D : are all points from the same patient #18 in replicates (n=7-11)? The description is unclear. For Figure 2D, arrows starting from “-Mo” point and ending to “+Mo” point could make it easier to highlight the message. In figure 4D, the statistics are presented from comparison to M-CSF-treated cells, not to untreated cells (first column), why?
- Figure 6D is not cited in the manuscript before figures 7-8 and should be moved as a new Figure 9. Please reorder figures accordingly to the manuscript organization. The results could be discussed in relation to data from Figure 7B. This new figure 9 could be completed with the data showing the dose-dependent induction of apoptosis under TNFa treatment that is introduced as “not shown” (lane534). For apoptosis assays, only U937 ell line is used. Why ? the other cell models could also be evaluated for comparison.
- The discussion is very long and could be reduced. Some part may be introduced in the results section and the discussion reduced, particularly to avoid some over-conclusions.
Minor points:
- The authors used several pharmacological drugs which mechanism of action / target should be indicated along the manuscript to facilitate the reading of the manuscript and clarity the message. In material and method section, please indicate the provider for each molecule that were used (or in supp Table together with antibodies for instance).
- For the list of antibodies, the precise epitope that is targeted should be indicated, especially against cytokines.
- MV-4-11 cell line is described and used as an AML cell line model for FLT3-ITD mutation, however, this cell line also present MLL-AF4 translocation as the major alteration leading to AML. Contrasting with Molm 13 cell line having the FL3-ITD in only one allele, the MV4-11 cell line present ITD in both alleles. This should be exemplified.
- In material and method section, what means “plain medium”? please specify.
- Lane 170: “GM-CSF+GW2580” and other places: how much?
- Statistics: “paired or unpaired t-test or Wilcoxon matched-pairs…”: it is unclear which one is used along the manuscript, please specify in the figure legends. Only for Figure 2A? and what is the rational to used paired tests for one or another experiments?
- There is a lot of references (144 references), maybe too much relatively to the standards of publications in Cancers journal. Please remove unnecessary references. Reference 37 should be completed.
- By contrast, some references may be interested to be discussed such as :
- Chen C, Wang R, Feng W, Yang F, Wang L, Yang X, Ren L, Zheng G. Peritoneal resident macrophages in mice with MLL-AF9-induced acute myeloid leukemia show an M2-like phenotype. Ann Transl Med. 2021 Feb;9(3):266. doi: 10.21037/atm-21-139.
- Neaga A, Bagacean C, Tempescul A, Jimbu L, Mesaros O, Blag C, Tomuleasa C, Bocsan C, Gaman M, Zdrenghea M. MicroRNAs Associated With a Good Prognosis of Acute Myeloid Leukemia and Their Effect on Macrophage Polarization. Front Immunol. 2021 Jan 15;11:582915. doi: 10.3389/fimmu.2020.582915. eCollection 2020.
Round 2
Reviewer 1 Report
The authors responded to all my comments.
Reviewer 2 Report
The present version is now improved and the proposed corrections were introduced along the manuscript.
I have no more request.